# Genetic Analysis of *SCN11A*, *SCN10A*, and *SCN9A* in Familial Episodic Pain Syndrome (FEPS) in Japan and Proposal of Clinical Diagnostic Criteria

**DOI:** 10.3390/ijms25136832

**Published:** 2024-06-21

**Authors:** Atsuko Noguchi, Tohru Tezuka, Hiroko Okuda, Hatasu Kobayashi, Kouji H. Harada, Takeshi Yoshida, Shinji Akioka, Keiko Wada, Aya Takeya, Risako Kabata-Murasawa, Daiki Kondo, Ken Ishikawa, Takeshi Asano, Michimasa Fujiwara, Nozomi Hishikawa, Tomoyuki Mizukami, Toshiaki Hitomi, Shohab Youssefian, Yoshihiro Nagai, Manabu Tanaka, Kaoru Eto, Hideaki Shiraishi, Fumimasa Amaya, Akio Koizumi, Tsutomu Takahashi

**Affiliations:** 1Department of Pediatrics, Akita University Graduate School of Medicine, 1-1-1 Hondo, Akita 010-8543, Japan; tomy@med.akita-u.ac.jp; 2Department of Pain Pharmacogenetics, Graduate School of Medicine, Kyoto University, Yoshida Konoe-cho, Sakyo-ku, Kyoto 606-8501, Japan; tezuka.toru.8w@kyoto-u.ac.jp (T.T.); hiroko.okuda@marianna-u.ac.jp (H.O.); aya_takeya@okamoto-hp.or.jp (A.T.); youssefian.shohab.55h@st.kyoto-u.ac.jp (S.Y.); akiokoizumi52@gmail.com (A.K.); 3Laboratory of Integrative Molecular Medicine, Graduate School of Medicine, Kyoto University, Yoshida Konoe-cho, Sakyo-ku, Kyoto 606-8501, Japan; 4Department of Preventive Medicine, St. Marianna University School of Medicine, 2-16-1, Sugao, Miyamae-ku, Kawasaki 216-8511, Japan; thitomi@marianna-u.ac.jp; 5Department of Environmental and Molecular Medicine, Mie University Graduate School of Medicine, Edobashi 2-174, Tsu 514-8507, Japan; hatasuk@med.mie-u.ac.jp; 6Department of Health and Environmental Sciences, Graduate School of Medicine, Kyoto University, Yoshida Konoe-cho, Sakyo-ku, Kyoto 606-8501, Japan; harada.koji.3w@kyoto-u.ac.jp; 7Department of Pediatrics, Kyoto University Graduate School of Medicine, 54 Shogoin Kawahara-cho, Sakyo-ku, Kyoto 606-8507, Japan; tayoshi@kuhp.kyoto-u.ac.jp; 8Department of Pediatrics, Graduate School of Medical Science, Kyoto Prefectural University of Medicine, 465 Kajii-cho, Kawaramachi-Hirokoji, Kamigyo-ku, Kyoto 602-8566, Japan; sakioka@koto.kpu-m.ac.jp; 9Department of Epidemiology and Preventive Medicine, Gifu University Graduate School of Medicine, 1-1 Yanagido, Gifu 501-1194, Japan; wada.keiko.z3@f.gifu-u.ac.jp; 10Department of Gynecology, Kyoto Okamoto Memorial Hospital, 100 Sayamanishi-No-Kuchi, Kumiyama-cho, Kuse-gun, Kyoto 613-0034, Japan; 11Department of Psychiatry, Iwate Prefectural Nanko Hospital, 17 Ohira, Kitsunezenji, Ichinoseki-shi 027-0031, Japan; risako-murasawa@pref.iwate.jp; 12Devision of Pediatrics, Akita Kousei Medical Center, 1-1-1 Iijima Nishibukuro, Akita 011-0948, Japan; kondod1199@akikumihsp.com; 13Department of Pediatrics, Iwate Medical University, 1-1 Iidai-dori 2-Chome, Yahaba-cho, Shiwa-gun 028-3695, Japan; 14Department of Pediatrics, Nippon Medical School Chiba Hokusoh Hospital, 1715 Kamagari, Inzai 270-1694, Japan; july1364@nms.ac.jp; 15Department of Pediatrics, NHO Fukuyama Medical Center, 14-17, 4-Chome, Okinogami-cho, Fukuyama City 720-8520, Japan; fujiwara.michimasa.kw@mail.hosp.go.jp; 16Department of Neurology, Kurashiki Heisei Hospital, 4-3-38 Oimatsu-cho, Kurashiki City 710-0826, Japan; nozomih.24@gmail.com; 17Department of Pediatrics, National Hospital Organization Kumamoto Medical Center, 1-5 Ninomaru, Chuo-ku, Kumamoto 860-0008, Japan; 18Laboratory of Molecular Biosciences, Graduate School of Medicine, Kyoto University, Yoshida Konoe-cho, Sakyo-ku, Kyoto 606-8501, Japan; 19Department of Pain Management and Palliative Care Medicine, Kyoto Prefectural University of Medicine, 465 Kajii-cho, Kawaramachi-Hirokoji, Kamigyo-ku, Kyoto 602-8566, Japan; navy@koto.kpu-m.ac.jp (Y.N.); ama@koto.kpu-m.ac.jp (F.A.); 20Division of General Pediatrics, Saitama Prefectural Children’s Medical Center, 1-2 Shin-Toshin, Chuo-ku, Saitama 330-8777, Japan; tanaka.manabu@saitama-pho.jp; 21Department of Pediatrics, Tokyo Women’s Medical University, 8-1 Kawada-cho, Shinjuku-ku, Tokyo 162-8666, Japan; eto.kaoru@twmu.ac.jp; 22Department of Pediatrics, Hokkaido University Hospital, North 15, West 7, Kita-ku, Sapporo 060-8638, Japan; siraisi@med.hokudai.ac.jp; 23Institute of Public Health and Welfare Research, 18-13 Uzumasa Tanamoricho, Ukyo-ku, Kyoto 616-8141, Japan

**Keywords:** familial episodic pain syndrome, *SCN11A*, *SCN10A*, *SCN9A*

## Abstract

Familial episodic pain syndrome (FEPS) is an early childhood onset disorder of severe episodic limb pain caused mainly by pathogenic variants of *SCN11A*, *SCN10A*, and *SCN9A*, which encode three voltage-gated sodium channels (VGSCs) expressed as key determinants of nociceptor excitability in primary sensory neurons. There may still be many undiagnosed patients with FEPS. A better understanding of the associated pathogenesis, epidemiology, and clinical characteristics is needed to provide appropriate diagnosis and care. For this study, nationwide recruitment of Japanese patients was conducted using provisional clinical diagnostic criteria, followed by genetic testing for *SCN11A*, *SCN10A*, and *SCN9A*. In the cohort of 212 recruited patients, genetic testing revealed that 64 patients (30.2%) harbored pathogenic or likely pathogenic variants of these genes, consisting of 42 (19.8%), 14 (6.60%), and 8 (3.77%) patients with variants of *SCN11A*, *SCN10A*, and *SCN9A*, respectively. Meanwhile, the proportions of patients meeting the tentative clinical criteria were 89.1%, 52.0%, and 54.5% among patients with pathogenic or likely pathogenic variants of each of the three genes, suggesting the validity of these clinical criteria, especially for patients with *SCN11A* variants. These clinical diagnostic criteria of FEPS will accelerate the recruitment of patients with underlying pathogenic variants who are unexpectedly prevalent in Japan.

## 1. Introduction

Pain can be classified as nociceptive, neuropathic, or nociplastic [1,2]. Nociceptors are specific primary sensory neurons that innervate tissues, including skin and muscle. Nociceptive pain involves nociceptors detecting noxious stimuli and releasing pain-modulating signals that are conducted by means of action potentials (APs) on sensory neurons, which propagate via the spinal cord to the central nervous system (CNS). In contrast, neuropathic pain is caused by a lesion or disease of the somatosensory nervous system as well as sensitization within the CNS.

Nociceptors express G protein-coupled receptors, nociceptive stimuli-transducing ligand-gated ion channels—including several members of the transient receptor potential (TRP) family—and a group of voltage-gated sodium channels (VGSCs), which are key determinants of excitability through integrating all-or-none APs and the propagation of APs to the CNS [1,2]. VGSCs comprise a pore-forming α subunit, which is associated with auxiliary β subunits. Among the nine different VGSCs (Nav1.1-Nav1.9) expressed in humans, Nav1.9, Nav1.8, and Nav1.7, encoded by *SCN11A*, *SCN10A*, and *SCN9A*, respectively, have been genetically and functionally validated as drivers of chronic pain in humans [2,3]. The biophysical properties of the three VGSCs determine nociceptor excitability; therefore, a gain-of-function or changed expression can lead to neuropathic pain. Over the last two decades, advances in genetic testing have revealed that Nav1.9, Nav1.8, and Nav1.7 are associated with neuropathic pain disorders, including familial episodic pain syndrome (FEPS), small fiber neuropathy, paroxysmal extreme pain disorder, primary erythromelalgia, and congenital insensitivity to pain [2].

Familial episodic pain syndrome is an autosomal dominant inherited disorder characterized by an early childhood onset of severe episodic pain that affects the distal extremities and tends to attenuate or diminish with age [1,4]. To date, FEPS has been subdivided into four variants associated with heterozygous pathogenic variants of *TRPA1*, *SCN11A*, *SCN10A*, and *SCN9A*, all of which cause functional gain-of-function effects. In 2010, a large Colombian family had 21 affected members with FEPS over four generations. Genetic screening revealed the presence of a heterozygous pathogenic variant of *TRPA1*, designated as FEPS type 1 [5]. However, no other FEPS type 1 families have been reported. Meanwhile, three FEPS type 2 patients were reported from two independent families, with atypical episodes of pain caused by two heterozygous pathogenic variants of *SCN10A* [6]. To date, over two dozen patients with early childhood onset FEPS have been associated with *SCN11A* (designated as FEPS type 3) [7,8,9,10,11,12,13,14]. FEPS type 3 patients are distributed among several races, including the Chinese, European, South American, and Japanese populations. Finally, patients of a Japanese family were reported to show episodic pain attacks from early childhood, which were revealed to be caused by a pathogenic variant of *SCN9A*, recently designated as FEPS type 4 [15]. Collectively, these three types of VGSCs may be the major etiologies of FEPS.

FEPS has been characterized by recurrent pain as an only symptom without any abnormalities of clinical laboratory tests or imaging studies [8,10]. Therefore, it might be challenging for medical practitioners to make a clinical diagnosis, leading to the presence of undetected patients. Therefore, nationwide survey and genetic testing are worth conducting to obtain epidemiological information on this disease. We considered it necessary to extend the analysis of these three genes to a large cohort of patients with the FEPS phenotype.

In addition, clinical diagnostic criteria for FEPS should be established to raise awareness of congenital pain syndromes in patients for medical practitioners, families, and even schoolteachers managing these patients [16]. Establishing clinical diagnostic criteria would also help to introduce patients to appropriate treatment and would provide a perspective for patients and clinicians.

For this study, we conducted nationwide recruitment of patients using the authors’ proposed clinical diagnostic criteria for FEPS through genetic testing for *SCN11A*, *SCN10A*, and *SNC9A*, and we established clinical diagnostic criteria for FEPS through the delineation of clinical features of the patients based on genetic testing.

## 2. Results

### 2.1. Patient Characteristics

As described in the Materials and Methods Section 4.3, we first outlined the clinical diagnostic criteria for FEPS that served as the basis for patient recruitment in this study. The diagnostic criteria consisted of major and minor criteria listed in Table 1. In this study, patients who met the following criteria were eligible to participate as FEPS patients: if two or more primary criteria and two or more secondary criteria were met, and there were no other diseases that could cause pain.

In total, 213 unrelated individuals were recruited for this study. This cohort included 6 and 42 individuals, for a total of 48, described by Okuda [8] and Kabata [10], respectively. One individual was excluded owing to Fabry’s disease. Eventually, 212 unrelated individuals (131 females, 61.8%) were analyzed for genetic variations in *SCN11A*, *SCN10A*, and *SCN9A* (Figure 1). The mean age of the patients at recruitment for this study was 10.5 ± 8.9 years, and the mean age of the onset of complaint was 3.1 ± 2.3 years.

### 2.2. SCN11A Variants Identified in FEPS Patients

In our cohort of 212 unrelated FEPS patients, exome sequencing revealed 14 variants in 46 patients (21.7%, *n* = 46/212) (Figure 1, Table 2). The 14 variants were scored according to the American College of Medical Genetics and Genomics (ACMG) standards and guidelines for the interpretation of sequence variants [17]. Five different variants—c.665G>A (p.R222H), c.664C>A (p.R222S), c.673C>T (p.R225C), c.2431C>T (p.L811F), and c.3437T>C (p.F1146S)—were scored as “pathogenic” variants. Seven variants—c.598T>C (p.S200P), c.923C>T (p.P308L), c.2410A>G (p.N804D), c.2441T>G (p.F814C), c.3551T>C (p.V1184A), c.4229C>T (p.T1410M), and c.4699T>C (p.C1567R)—were identified as “likely pathogenic” variants. Variant c.5309T>C (p.L1770P) was scored as a variant of unknown significance (VUS). Finally, the variant c.2669G>A (p.G890G) was scored as “likely benign” (Figure 2a). Among these, five types of variants were observed in one or more patients. Notably, a pathogenic variant—c.665G>A (p.R222H)—was identified in 25 individuals in this cohort (11.8%, 25/212), accounting for a relatively large proportion.

### 2.3. SCN10A Variants Identified in FEPS Patients

In our cohort of 212 unrelated FEPS patients, exome sequencing revealed 18 heterozygous variants in 25 patients (11.8%; 25/212) (Figure 1, Table 2). Eighteen different variants were scored according to the ACMG standards and guidelines for the interpretation of sequence variants [17]. The variant p.I1708del was scored as “pathogenic”. Nine different variants—c.53C>T (p.P18L), c.565T>A (p.W189R), c.1157T>G (p.F386C), c.1489C>T (p.R497C), c.2161C>T (p.P721S), c.2311C>T (p.P771S), c.3670C>T (p.L1224F), c.4379G>C (p.R1460P), and c.5605C>T (p.R1869C)—were scored as “likely pathogenic”. Five variants—c.110C>T (p.A37V), c.4580T>C (p.M1527T), c.5047C>T (p.P1683S), c.5536C>A (p.L1846I), and c.5831A>T (p.D1944V)—were variants of VUS. Three variants—c.349A>C (p.N117H), c.1277G>A (p.R426Q), and c.1549C>T (p.P517S)—were identified as “likely benign” (Figure 2b).

### 2.4. SCN9A Variants Identified in FEPS Patients

In our cohort of 212 unrelated FEPS patients, exome sequencing revealed eight different heterozygous variants in 11 patients (5.2%; 11/212) (Figure 1, Table 2). The eight variants were scored according to the ACMG standards and guidelines for the interpretation of sequence variants [17]. Two variants—c.29A>G (p.Q10R) and c.130G>C (p.E44Q)—were classified as “pathogenic”. Three variants—c.3335G>A (p.S1112N), c.4739T>C (p.V1580A), and c.5678G>A (p.R1893H)—were classified as “likely pathogenic”. Two variants—c.3179T>C (p.M1060T) and c.3312C>A (p.S1104R)—were classified as VUS. The variant c.1818T>G (p.S606R) was scored as a “likely benign” variant (Figure 2c).

### 2.5. Clinical Features of FEPS Patients with SCN11A, SCN10A, or SCN9A Variants

To delineate the clinical features of FEPS, the patients were analyzed in three subgroups, with pathogenic or likely pathogenic variants of *SCN11A*, *SCN10A*, and *SCN9A* (Table 3). Six subjective symptoms concerning the nature of pain attacks, mean age at the onset of pain attacks, mean frequency of pain attacks in a month, intermittency of pain in an attack, periodicity of pain in an attack, mean frequency of periodic pain in an attack, and local position of pain were characterized in all cohorts and subgroups. The mean age of onset of pain attacks in the subgroup with pathogenic or likely pathogenic variants of *SCN11A* was significantly younger than that in the subgroup with pathogenic or likely pathogenic variants of *SCN10A* (1.74 ± 1.1 vs. 3.20 ± 1.50, *p* = 0.001) or *SCN9A* (1.74 ± 1.1 vs. 4.27 ± 2.8; *p* = 0.037) (Table 3, Figure 3). Furthermore, a significant difference was observed between *SCN11A* and *SCN10A* in terms of pain intermittency during attacks (*p* < 0.001). No significant differences were observed in the other subjects concerning the nature of the pain attacks among the subgroups.

Four frequent complaints or complications—migraine, gastrointestinal symptoms, muscle symptom, and feeling of coldness in limbs—were characterized in all cohorts and subgroups. Migraine, gastrointestinal symptoms, muscle symptoms, and feelings of coldness were observed in 31.3% (10/32), 39.4% (13/33), 56.7% (17/30), and 32.5% (13/40), respectively, of the subgroups with pathogenic or likely pathogenic variants of *SCN11A*. There were no significant differences in the frequencies among the subgroups. From the questionnaire in this study, many patients used acetaminophen and non-steroidal anti-inflammatory drugs (NSAIDS, mainly ibuprofen) as medications for pain attacks; however, the efficacy of the drugs could not be evaluated in this study.

### 2.6. Validating Clinical Diagnostic Criteria in Individuals with Possible FEPS

For patient recruitment, we used tentative clinical diagnostic criteria (Table 1), which are based on the clinical features of previously reported patients [8,10]. Each of the three primary criteria (A, recurrent episodic pain beginning in early childhood; B, pain episodes occurring primarily in the limbs; and C, pain episodes occurring three or more times a month) were recognized in 86.7% (184/212), 91.9% (195/212), and 87.7% (186/212) of the cohorts, respectively. No significant differences were seen for the clinical findings of the three primary criteria in any of three subgroups. Each of the three secondary criteria [(1) presence of family history (note: pain attacks often subside in adulthood but may persist in some cases); (2) episodes triggered by cold temperatures, low atmospheric pressure, bad weather, or fatigue; and (3) excruciating pain that interferes with daily living and sleep] were recognized in 76.4% (162/212), 75.0% (159/212), and 81.1% (172/212) of the cohort, respectively. No significant differences were seen for the clinical findings of the three secondary criteria in the three subgroups.

Definite FEPS was diagnosed when all three primary criteria were met, and the patient harbored a pathogenic variant of *SCN11A*, *SCN10A*, or *SCN9A* without other diseases that may cause pain. A probable diagnosis of FEPS was made when all three primary criteria were met with a family history (Secondary Criterion 1) and two or more secondary criteria without other diseases that may cause pain.

To validate the concordance between the genetic findings and clinical diagnostic criteria, we examined the proportion of the patients who met the three primary criteria (A, B, and C) by the class of the found *SCN9A*, *10A*, and *11A* variants, which is based on ACMG standards and guidelines for the interpretation of sequence variants [17]. In this paper, we classified pathogenic or likely pathogenic variants as Class I, VUS variants as Class II, and benign or likely benign variants as Class III, conveniently (Table 4). In 46 patients of the *SCN11A* variant group, 42 patients harbored the pathogenic or likely pathogenic variants (Class I in Table 4) of *SCN11A*. Of the 42 patients, 41 (97.6%) met the primary criteria, which were compatible with the definite criteria. In 25 patients of the *SCN10A* variant group, 14 patients harbored the Class I variant. Of the 14 patients, 13 (92.6%) met the primary criteria, which were compatible with the definite criteria. In the 11 patients of the *SCN9A* variant group, 8 patients harbored the Class I variant. In total, six of the eight patients (75.0%) met the primary criteria, compatible with the definite criteria. Out of the total of 82 patients, 60 harbored the Class I variant of *SCN11A*, *SCN10A*, or *SCN9A*, compatible to the primary criteria, resulting in a positivity rate of 73.2%. The diagnostic criteria showed a high sensitivity of 93.7% in all groups.

In contrast, 4 of the 46 patients in the *SCN11A* variant group harbored VUS, benign, or likely benign variants (Class II or III). Out of the four patients, two (50.0%) met the probable criteria. Similarly, among the 25 patients in the *SCN10A* variant group, 11 harbored a Class II or III variant. In the 11 patients, 6 patients (54.5%) met the probable criteria. In 11 patients of the *SCN9A* variant group, 3 patients harbored a Class II or III variant. Out of the three patients, two (66.7%) met the probable criteria. In total, 18 patients harbored a Class II or III variant of the three genes. Out of them, 10 (6 + 4: 56%) were compatible with the probable criteria.

## 3. Discussion

In this study, patients who complained of episodic pain attacks along with the previously described phenotype of FEPS were widely recruited and analyzed by genetic testing of *SCN11A*, *SCN10A*, and *SCN9A*, followed by evaluation of the genetic variants according to the ACMG standards and guidelines for the interpretation of sequence variants [17]. In the cohort of 212 unrelated FEPS patients, 64 (30.2%) were diagnosed with FEPS based on the presence of pathogenic or likely pathogenic variants of *SCN11A*, *SCN10A*, or *SCN9A*. Furthermore, 10 patients (4.7%) harbored VUS of *SCN11A*, *SCN10A*, or *SCN9A* (Figure 1). When including VUS, a total of 74 patients (34.9%) presented with the FEPS phenotype owing to genetic variants of the three genes.

As for FEPS type 1, *TRPA1* was evaluated only in the 154 patients analyzed by whole-exome sequencing in our cohort, leading to the identification of three variants with uncertain significance.

This indicates that these three genes play major roles in the etiology in a clinical entity of FEPS, while also suggesting other genes may be responsible for FEPS.

During pain generation, Nav1.7–1.9 play a key role of the firing in primary sensory neurons [2]. In VGSCs, the Nav1.9 channel is responsible for amplifying subthreshold stimuli [2]. The Nav1.7 channel is also responsible for amplifying the subthreshold and is involved in the formation of the action potential ramus. Therefore, both Nav1.7 and Nav1.9 are involved in the summation of subthreshold stimuli and determine the threshold for action potential generation. In contrast, the Nav1.8 channel is the main VGSC responsible for the formation of the action potential rising phase [2].

In our study, 85% of the Nav1.9 variants in FEPS were classified into Class I and almost all of them were located in the voltage censer domain and the pore-forming region (Figure 2a). Previously, three of these variants, p.R222S, p.F814C, and p.F1146S, were functionally analyzed using the dorsal root ganglia (DRG) neurons of the genetic model mice, showing the hyperexcitability of the neurons caused by these variants [8,10]. Therefore, it could be speculated that these genetic variants, which have not yet been functionally analyzed, also result in changes in channel function, leading to the hyperexcitation of neurons. Ten Nav1.8 variants of Class I have been found in FEPS in this study and all of those are located in the transmembrane (Figure 2b), and some of them have been reported as genetic variants related to Brugada syndrome [18,19]. However, there were no such patients with Brugada syndrome or cardiac disease in our cohort. As shown in Figure 2c, most of the Nav1.7 variants in FEPS were located in the linker regions of the channel. Specifically, two Nav1.7 pathogenic variants of p.Q10R and p.E44Q were located in the N-terminal region. p.Q10R and p.E44Q were previously investigated on the channel property and p.Q10R, which was identified from the patient of erythromelalgia, showed a hyperpolarizing shift in channel activation [15,20]. The channel properties of the other Nav1.7 variants have not been investigated thus far.

There were five variants of *SCN11A* in multiple different individuals of FEPS. Surprisingly, the p.R222H variant accounted for 11.8% of patients in this cohort. The p.R222H variant was first reported by Okuda et al., who identified p.R222H in five Japanese families with FEPS. Since then, other families with FEPS caused by the p.R222H variant were reported in South American, Chinese, European, and Japanese populations, suggesting the recurrence of the p.R222H variant as an etiology of FEPS [8,9,11,12]. Two pathogenic variants [c.673C>T (p.R225C) and c.3437T>C (p.F1146S)] were detected in three and six individuals, respectively.

The clinical features of FEPS have been studied in patients with pathogenic or likely pathogenic variants of *SCN11A*, *SCN10A*, and *SCN9A*. Characteristically, the mean age at onset of pain attacks in the subgroup with Class I variants of *SCN11A* is significantly lower than that in the subgroup with pathogenic or likely pathogenic variants of *SCN10A* or *SCN9A*. In FEPS patients, episodic pain attacks appear in an age-dependent manner and are characterized by an early childhood onset of episodic pain that mainly affects the distal extremities and tends to attenuate or diminish with age. However, the reason for the age-dependent appearance of pain attacks remains unknown. This suggests that there are developmental changes in VGSC function or the sensation of pain in humans. A few experimental observations of gene expression and protein glycosylation have suggested functional changes in VGSCs from neonates to adults. Neonatal rat DRG membrane contained more extensive glycosylation of Nav1.9 compared with adult DRG neurons and the developmental change in the glycosylation state of Nav1.9 is paralleled by a developmental change in the gating of the persistent Na^+^ current attributable to Nav1.9 in native DRG neurons [21]. In another report, the developmental expression of Nav1.9 was studied in rat DRG neurons, demonstrating that the expression of Nav1.9 channels peaks in adolescence and significantly declines after the beginning of adulthood [22]. These experimental observations may be related to the age-dependent appearance of pain attacks, while, in our previous report, the aged mice (36–38 weeks of age) which harbored one of the variants of *SCN11A* p.R222S did not alleviate the pain symptoms [8]. Further research is needed for a detailed understanding of the related pathogenesis.

In this study, we propose a clinical diagnostic criteria for FEPS to provide a perspective for clinicians who assess patients with recurrent pain attacks. The clinical criteria were validated in terms of the relationship between positive pathogenic or likely pathogenic variants and the fulfillment of the proposed diagnostic clinical criteria.

There was no difference in clinical features between the groups with VGSC variants and the group without a VGSC variant. Therefore, it is challenging to determine whether a patient has a pathogenic VGSC variant solely based on these clinical diagnostic criteria. However, most patients with a pathogenic VGSC variant meet these clinical criteria, indicating that these criteria encompass the clinical picture of patients with pathogenic VGSC variants. Consequently, the authors speculate that while these criteria do not necessarily indicate that a patient has a pathogenic VGSC variant, they are reasonably effective in identifying patients who might have a pathogenic VGSC variant. This is the first set of clinical diagnostic criteria for FEPS, providing a diagnostic pathway for FEPS patients.

Our study has some limitations. The utilization of genetic testing showed some limitations for the definitive diagnosis of FEPS, due to the inaccessibility of functional analyses of the VGSC VUS. In this study, 10 patients (4.7%) had VUS of *SCN11A*, *SCN10A*, or *SCN9A*. Some VUS will be evaluated as novel pathological variants through the accumulation of more patient studies. And there may be some bias in the diagnostic criteria related to the *SCN11A* phenotype since we developed diagnostic criteria based on *SCN11A* FEPS. In fact, there is heterogeneity in the age of onset among patients; patients with *SCN11A*-FEPS developed pain symptoms younger than other patients. However, the clinical diagnostic criteria proposed in this study were significant and helpful for diagnosis in combination with genetic testing and clinical diagnostic criteria.

## 4. Materials and Methods

### 4.1. Ethical Statement

Clinical and genetic studies on humans were approved by the Institutional Review Board and Ethics Committee of Kyoto University School of Medicine, Japan (Approval No. G501; approval date, 2 August 2012), and Akita University Graduate School of Medicine, Japan (Approval No. 960; approval date, 26 September 2012). Written informed consent was obtained from all participants and the parents of the children and adolescents before participation.

### 4.2. Study Population (Study Design and Subjects)

Patients were recruited from hospitals in Japan, and both questionnaires and genetic testing were conducted from April 2015 to March 2023. From February 2020 to September 2021, 1597 hospitals that were statistically evenly extracted from 2604 hospitals registered as providing pediatric care in Japan were surveyed by mail regarding their experience in caring for FEPS patients. Consequently, 37 hospitals had FEPS patients out of a total reply from 993 hospitals (response rate: 62.2%). Subsequently, a questionnaire survey was conducted to collect detailed information from the 37 hospitals, and genetic testing was conducted if informed consent was obtained. Thereafter, until March 2023, patients were recruited via the website and medical conferences, and both questionnaires and genetic testing were conducted.

### 4.3. Inclusion/Exclusion Criteria and Definition of Familial Episodic Pain Syndrome

We proposed clinical diagnostic criteria for FEPS based on previously described patient reports [7,8,9,10,11,12]. This was drafted as an expert opinion based on past academic papers and clinical experiences, and serve as the basis for patient recruitment for this study. The diagnostic criteria were composed of primary and secondary criteria (Table 1).

For the primary criteria, we addressed the clinical features of pain episodes. Recurrent episodic pain begins during early childhood (Primary Criterion A). Pain episodes primarily occur in the limbs (Primary Criterion B) and episodes of attack occur three or more times a month (Primary Criterion C). In the secondary criteria, we addressed family history and inducing factors: (1) presence of family history (note: pain attacks often subside in adulthood but may persist in some cases); (2) episodes triggered by cold temperatures, low atmospheric pressure, bad weather, or fatigue; and (3) excruciating pain that interferes with daily living and sleep (Table 1).

In this study, patients who met the following criteria were eligible to participate as FEPS patients: if two or more primary criteria and two or more secondary criteria were met, and there were no other diseases that may cause pain.

To validate the primary criteria, patients were evaluated using genetic testing results. The rates of patients who met Primary Criteria A, B, and C were determined in each of the groups with *SCN11A*, *SCN10A*, and *SCN9A* variants of the pathogenic or likely pathogenic, VUS (variant of uncertain significance), and likely benign or benign classifications. Then, the patient was defined as definite FEPS when all three primary criteria were met, and the patient harbored a pathogenic variant of *SCN11A*, *SCN10A*, or *SCN9A* without other diseases that may cause pain, and the patient was defined as probable FEPS when all three primary criteria were met with a family history (Secondary Criterion 1) and two or more secondary criteria without other diseases that may cause pain (Table 4).

### 4.4. Survey Items

The following items were investigated for each patient with FEPS who was treated at the study hospital: age, sex, age of onset, painted point of the extremities (finger, wrist, forearm, elbow, upper arm, neck, shoulder, back, hip, thigh, knee, shin, calf, ankle, toes, and others), frequency of pain (monthly, daily, seasonal), pain triggers (cold, bad weather, low pressure, fatigue, others), method of mitigation, complications, and medications.

### 4.5. DNA Extraction and Storage

Genomic DNA was extracted from whole blood samples using a QIAamp DNA Blood Mini Kit (Qiagen, Hilden, Germany). DNA extraction was conducted according to the manufacturer’s protocol and the DNA stored at −20 °C.

### 4.6. Genetic Analysis of SCN11A, SCN10A, and SCN9A Genes

Whole-exome sequencing was performed on 125 probands at Riken Genesis Co., Ltd. (Tokyo, Japan) as previously described [8,10]. For the other 29 probands, whole-exome sequencing was performed at Kyoto University, as previously described [23]. Briefly, DNA was captured using the xGen Exosome Research panel kit (Integrated DNA Technologies, Coralville, IA, USA) and sequenced on a DNBSEQ-G400 (MGI Tech, Beijing, China) with 150 bp paired-end reads. Variants were identified using the Genomon2 pipeline (https://genomon.readthedocs.io/ja/latest/, accessed on 9 January 2023). For the remaining probands, all exons and intron–exon boundaries of *SCN11A*, *SCN10A*, and *SCN9A* were amplified by PCR using the primer sets listed in Appendix A and subjected to Sanger sequencing. Candidate variants were defined as non-synonymous variants with a minor allele frequency (MAF) of less than 0.01 in the Japanese population reference panel “38KJPN” (https://jmorp.megabank.tohoku.ac.jp/, accessed on 17 January 2024). Available family members were analyzed using Sanger sequencing for the presence of variants identified in the probands.

In this study, some patients harbored two or more variants in one or more genes of *SCN11A*, *SCN10A*, and *SCN9A*. If one of the variants was evaluated as pathogenic or likely pathogenic, the gene with the pathogenic or likely pathogenic variant was identified as the responsible gene. If two or more variants were evaluated as pathogenic or likely pathogenic, one of the three genes with the pathogenic or likely pathogenic variants was determined to be the responsible gene and was prioritized in the order of *SCN11A*, *SCN10A*, and *SCN9A*.

### 4.7. Variant Classification

FEPS patients were recruited and sorted according to the clinical diagnostic criteria described above, and variants of the *SCN11A*, *SCN10A*, and *SCN9A* genes were determined. First, we focused on the deleterious variations of the genes from the detected variants using the following series of filters: (1) MAF < 0.001 in the Japanese population from the ToMMo 38KJPN (https://jmorp.megabank.tohoku.ac.jp/, accessed on 17 January 2024); (2) validation of variants present in affected members and not present in unaffected members in each pedigree; and (3) non-synonymous variants (missense, nonsense, frameshift, and splice variants). We subsequently investigated whether the variants were known to be related to pain or other diseases or were novel variants causing pain. Genetic variants of *SCN11A*, *SCN10A*, and *SCN9A* were classified according to the ACMG standards and guidelines for the interpretation of sequence variants [17]. These guidelines provide criteria for classifying pathogenic variants into four categories. In particular, the four categories were subdivided into PS (strong pathogenicity), PM (moderate pathogenicity), and PP (supporting pathogenicity), and some of them were arranged to adapt to infantile pain syndrome as follows. PS2 and PM6: our variant data were mixed with multiple and/or single pedigree data; therefore, we subcategorized the number of pedigree data and performed parent analysis (for example, PS2 was considered ND if only one parent was analyzed, regardless of the presence or absence of the variant). PM1: the original category is “located in a mutational hot spot and/or critical and well-established functional domain (such as the active site of an enzyme) without benign variation” [19], while we defined four channel domains, pore regions, and four motifs (phosphorylation sites, IFM motif, IQ motif, and PY/PXY motif) [17] as functional sites corresponding to channel functions.

### 4.8. Statistical Analysis

Data were analyzed using the IBM SPSS statistics 28.0.0.0 (190) software package and the results are presented as the mean ± standard deviation. The Kruskal–Wallis test was used to compare the mean differences between the two groups. Comparisons of population proportions were analyzed using Pearson’s chi-square test. Statistical significance was set at *p* < 0.05.

## 5. Conclusions

Through nationwide recruitment in Japan, 212 unrelated FEPS patients were evaluated for genetic variants. Of them, 19.8%, 6.6%, and 3.8% harbored pathogenic or likely pathogenic variants of *SCN11A*, *SCN10A*, and *SCN9A*, respectively.

Of note, the accumulation of the c.665G>A (p.R222H) of *SCN11A* variant was implicated as the cause of FEPS in this cohort. Our data suggest that VGSC variants constitute a significant minority of the pathogenesis of FEPS. Among the three VGSC genes, *SCN11A* was the most common and the disease course was predominately typical, including an early age of onset. However, including VUSs, only 35% of FEPS patients had a variant in those three genes. Thus, in the majority of cases, the genetic predisposition of FEPS has not yet been established and there may be other genes commonly involved in this disorder.

Most patients with pathogenic or likely pathogenic variants of *SCN11A, SCN10A*, and *SCN9A* met the clinical diagnostic criteria proposed in this study. This suggests that the criteria may be useful for the detection of patients with the disease. In addition, more intensive studies in the future may identify other FEPS genes from our cohort.

## Figures and Tables

**Figure 1 ijms-25-06832-f001:**
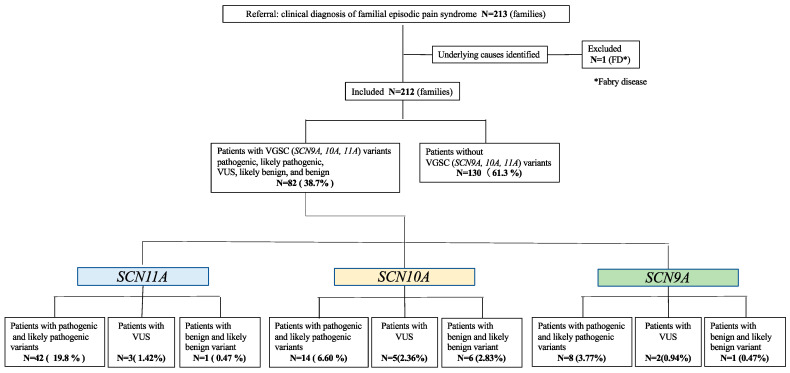
FEPS patients analyzed for variants of *SCN11A*, *SCN10A*, and *SCN9A*. The figure shows an overview of the study cohort. A total of 213 individuals were recruited, with one individual excluded under the diagnosis of Fabry’s disease. Screening for variants of three VGSCs (*SCN11A, SCN10A*, and *SCN9A*) was conducted for the remaining 212 individuals, revealing variants in 82 individuals. Among them, 42 individuals (19.8%) had pathogenic or likely pathogenic variants of *SCN11A*, 14 individuals (6.6%) had variants of *SCN10A*, and 8 individuals (3.77%) had variants of *SCN9A*.

**Figure 2 ijms-25-06832-f002:**
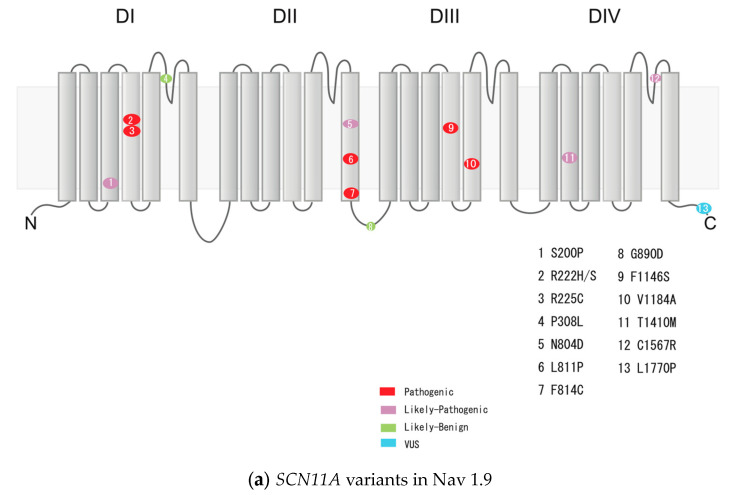
Genetic variants of *SCN11A*, *SCN10A*, and *SCN9A* identified in this study and their location in the Nav1.9, Nav1.8, and Nav1.7. Nav1.9, Nav1.8, and Nav1.7 have four domains (DI~DIV), each of which consistent of six transmembrane segments. Genetic variants are shown in schematic diagrams for Nav1.9 (**a**), 1,8 (**b**), and 1.7 (**c**), respectively. The color of each variant indicates its rating, based on the American College of Medical Genetics and Genomics standards and guidelines for interpreting sequence variants. Red represents “pathogenic,” purple represents “likely pathogenic,” green represents “likely benign,” and light blue represents “VUS”.

**Figure 3 ijms-25-06832-f003:**
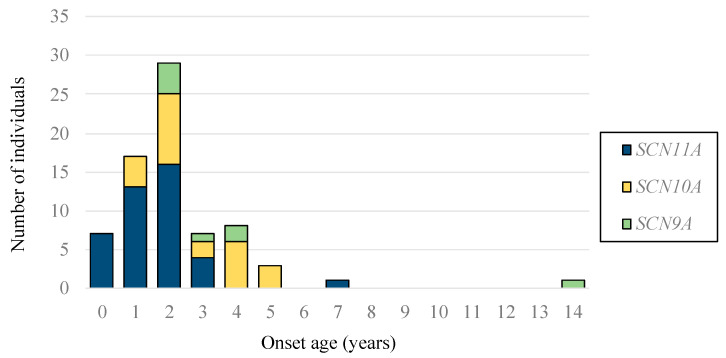
Onset age of pain attacks in FEPS patients with pathogenic or likely pathogenic variants of *SCN11A*, *SCN10A*, or *SCN9A*. The blue column indicates *SCN11A*, the yellow column indicates *SCN10A*, and the green column indicates *SCN9A*.

**Table 1 ijms-25-06832-t001:** Clinical diagnostic criteria for familial episodic pain syndrome (FEPS).

**Primary Criteria**
(A) recurrent episodic pain beginning in early childhood
(B) pain episodes occurring primarily in the limbs
(C) pain episodes occur three or more times a month
**Secondary Criteria**
(1) presence of family history (Note; pain attacks often subside in adulthood but may persist in some cases)
(2) episode can be triggered by cold temperatures, low atmospheric pressure, bad weather, or fatigue
(3) excruciating pain that interferes with activities of daily living and sleep

**Table 2 ijms-25-06832-t002:** Identification of gene variants of *SCN11A*, *SCN10A*, and *SCN9A* in a cohort of FEPS candidates.

Gene	c.Position	p.Position	Number of Patients	Location (GRCh38.p14)	ToMMo 38KJPN Allele Frequency	Variant Classification	dbSNP	* Reference
*SCN11A*	c.598T>C	p.S200P	1	chr3:38926822	na	likely pathogenic	na	
	c.665G>A	p.R222H	25	chr3:38925462	0.000013	pathogenic	rs1230622899	[8,9]
	c.664C>A	p.R222S	1	chr3:38925463	na	pathogenic	rs775199760	[8]
	c.673C>T	p.R225C	3	chr3:38925454	na	pathogenic	rs138607170	[10,13]
	c.923C>T	p.P308L	2	chr3:38919971	0.001846	likely pathogenic	rs751477540	
	c.2410A>G	p.N804D	1	chr3:38894958	na	likely pathogenic	na	
	c.2431C>T	p.L811F	1	chr3:38894937	na	pathogenic	na	
	c.2441T>G	p.F814C	1	chr3:38894927	na	likely pathogenic	na	[10]
	c.2669G>A	p.G890D	1	chr3:38894699	0.000581	likely benign	rs769754010	
	c.3437T>C	p.F1146S	5	chr3:38872251	na	pathogenic	na	[10]
	c.3551T>C	p.V1184A	1	chr3:38871653	na	likely pathogenic	na	[10,14]
	c.4229C>T	p.T1410M	1	chr3:38850579	0.000129	likely pathogenic	rs771241253	
	c.4699T>C	p.C1567R	1	chr3:38847371	0.002595	likely pathogenic	rs201595463	
	c.5309T>C	p.L1770P	3	chr3:38846761	0.005023	VUS	rs148328451	
*SCN10A*	c.53C>T	p.P18L	4	chr3:38793958	0.006521	likely pathogenic	rs190176472	
	c.110C>T	p.A37V	1	chr3:38793901	na	VUS	na	
	c.349A>C	p.N117H	2	chr3:38792090	0.000852	likely benign	rs774462243	
	c.565T>A	p.W189R	1	chr3:38771313	0.000245	likely pathogenic	rs1379282429	
	c.1157T>G	p.F386C	4	chr3:38756807	0.0043	likely pathogenic	rs78555408	
	c.1277G>A	p.R426Q	2	chr3:38756687	0.001485	likely benign	rs143033805	
	c.1489C>T	p.R497C	1	chr3:38752485	0.000077	likely pathogenic	rs370009920	
	c.1549C>T	p.P517S	2	chr3:38752425	0.000413	likely benign	rs2063758700	
	c.2161C>T	p.P721S	3	chr3:38739634	0.000723	likely pathogenic	rs747114420	
	c.2311C>T	p.P771S	1	chr3:38728871	0.000026	likely pathogenic	na	
	c.3670C>T	p.L1224F	2	chr3:38718664	0.001265	likely pathogenic	rs200597401	
	c.4379G>C	p.R1460P	1	chr3:38707286	na	likely pathogenic	rs369399424	
	c.4580T>C	p.M1527T	1	chr3:38701916	na	VUS	na	
	c.5047C>T	p.P1683S	1	chr3:38698173	0.006637	VUS	rs146999807	
	c.5122_5124del	p.I1708del	1	chr3:38698103_38698105	na	pathogenic	na	
	c.5536C>A	p.L1846I	3	chr3:38697684	0.001911	VUS	rs1001583386	
	c.5605C>T	p.R1869C	2	chr3:38697615	0.002247	likely pathogenic	rs141648641	
	c.5831A>T	p.D1944V	1	chr3:38697389	na	VUS	rs768502791	
*SCN9A*	c.29A>G	p.Q10R	2	chr2:166311728	0.001111	pathogenic	rs267607030	[16]
	c.130G>C	p.E44Q	1	chr2:166311627	0.000039	pathogenic	rs757848676	[13]
	c.1818T>G	p.S606R	3	chr2:166284609	0.004855	likely benign	rs202141567	
	c.3179T>C	p.M1060T	1	chr2:166272538	na	VUS	rs781223783	
	c.3312C>A	p.S1104R	1	chr2:166272405	0.000362	VUS	rs767904709	
	c.3335G>A	p.S1112N	3	chr2:166251869	0.006529	likely pathogenic	rs141040985	
	c.4739T>C	p.V1580A	1	chr2:166203957	na	likely pathogenic	rs1574705360	
	c.5678G>A	p.R1893H	3	chr2:166198928	0.007657	likely pathogenic	rs79805025	

na: not applicable. * The variant is previously reported.

**Table 3 ijms-25-06832-t003:** Clinical features of FEPS patients with pathogenic or likely pathogenic variants of *SCN11A*, *SCN10A*, or *SCN9A*.

	*SCN11A*-FEPS Pathogenic and Likely Pathogenic	*SCN10A*-FEPS Pathogenic and Likely Pathogenic	*SCN9A*-FEPS Pathogenic and Likely Pathogenic	Patient without *SCN9A*, *SCN10A*, or *SCN11A* Variant
	(n = 42)	(n = 14)	(n = 8)	(n = 130)
male:female	21:21	4:10	2:6	47:83
mean age at recruitment (range, median)	9.3 (range 0.08–71, median 6.3)	9.48 (range 0.1–18.1, median 11.1)	15.6 (range 5.0–53, median 11.1)	10.3 (range 2.16–53, median 8.9)
**History of primary criteria**				
A) recurrent episodic pain beginning in early childhood (n)	41	14	6	110
B) pain episodes occurring primarily in the limbs (n)	42	14	8	115
C) pain episodes occur more than several times a month (n)	42	13	7	110
**History of secondary criteria**				
a) presence of family history (n)	40	10	5	94
b) episode can be triggered by cold temperatures, low atmospheric pressure, bad weather or fatigue (n)	36	10	6	94
c) excruciating pain that interferes with activities of daily living and sleep (n)	38	14	5	99
**Nature of pain attack**				
mean age of onset of pain attack (years ± SD)	1.74 ± 1.1 (range 0–6.0) ^a^	3.20 ± 1.50 (range 1.5–5.0) *	4.27 ± 2.8 (range 2.0–14.7) *	3.21 ± 2.1
mean frequency of pain attack in a month (n ± SD)	13.0 ± 7.1 (range 3–30)	11.7 ± 6.8 (range 2–20)	14.2 ± 7.9 (range 2–30)	14.5 ± 9.3
intermittency of pain in an attack (n)	38/40 ^a,b^	6/13 ^b,^*	4/6 ^b^	70/115 ^b^
periodicity of pain in an attack (n)	1/39 ^b^	0/12 ^b^	0/5 ^b^	7/110 ^b^
mean frequency of periodic pain in an attack (n ± SD)	4.7 ± 1.9 (range 2–7)	5.3 ± 5.3 (range 2–10)	3.8 ± 1.2 (range 2–5)	3.4 ± 2.7
nature of pain attack; local position of pain				
finger (n), wrist (n), forearm (n), elbow (n), upper arm (n), shoulder (n), neck (n), back (n), hip jount (n), thigh (n), knee (n), lower leg (n), ankle (n), toe (n)	finger (6), wrist (17), forearm (8), elbow (12), upper arm (6), shoulder (1), neck (0), back (1), hip joint (2), thigh (11), knee (32), lower leg (19), ankle (18), toe (12)	finger (0), wrist (9), forearm (3), elbow (4), upper arm (1,) shoulder (0), neck (0), back (0), hip jount (2), thigh (4), knee (7), lower leg (10), ankle (8), toe (2)	finger (1), wrist (3), forearm (5), elbow (3), upper arm (3), shoulder (0), neck (0), back (0), hip jount (1), thigh (3), knee (5), lower leg (9), ankle (3), toe (1)	finger (19), wrist (48), forearm (35), elbow (39), upper arm (11), shoulder (13), neck (11), back (6), hip joint (24), thigh (40), knee (82), lower leg (111), ankle (68), toe (18)
migraine (n)	10/32 ^b^	4/13 ^b^	2/6 ^b^	36/111 ^b^
gastrointestinal symptoms (n)	13/33 ^b^	2/13 ^b^	1/6 ^b^	31/111 ^b^
muscle symptoms (n)	17/30 ^b^	5/12 ^b^	3/7 ^b^	43/102 ^b^
feel of coldness in limb (n)	13/40 ^b^	1/12 ^b^	2/7 ^b^	30/107 ^b^

^a^ There is a significant difference from *, ^b^ The population is different because some patients did not answer their information.

**Table 4 ijms-25-06832-t004:** Relationship between variant classification and primary criteria for FEPS.

			FEPS Criteria									
			With Three Primay Criteria						Without Three Primary Criteria			
Gene	N	Variant Class	Definitive	%	Probable	%	Total (a)	%		%	Total (b)	(a/b) (%)
*SCN11A*	46	I	41	89.1	-	-	41	89.1	1	2.17	42	97.6
		II	-	-	1	2.17	1	2.17	2	4.3	3	33.3
		III	-	-	1	2.17	1	2.17	0	0	1	100
		Subtotal	41	89.1	2	4.3	43	93.5	3	6.5	46	93.5
*SCN10A*	25	I	13	52	-	-	13	52	1	4	14	92.6
		II	-	-	4	16	4	16	1	4	5	80
		III	-	-	2	8	2	8	4	16	6	33
		Subtotal	13	52	6	24	19	76	6	24	25	76
*SCN9A*	11	I	6	54.5	-	-	6	54.5	2	18.2	8	75
		II	-	-	1	9.1	1	9.1	1	9.1	2	50
		III	-	-	1	9.1	1	9.1	0	0	1	100
		Subtotal	6	54.5	2	18.2	8	72.7	3	27.3	11	72.7
Patient with *SCN9A* or *SCN10A* or *SCN11A* variant	82	I	60	73.2	-	-	60	73.2	4	4.88	64	93.7
		II	-	-	6	7.4	6	7.4	4	4.88	10	60
		III	-	-	4	4.94	4	4.93	4	4.88	8	50
		Total	60	73.2	10	12.2	70	85.3	12	14.6	82	85.4

I: Pathogenic/Likely pathogenic, II: VUS, III: Benign/Likely benign.

## Data Availability

The original contributions presented in the study are included in the main text/Appendix A. Alternatively, the data presented in this study are available on request from the corresponding author.

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
