# Peer review of "Genetic Analysis of SCN11A, SCN10A, and SCN9A in Familial Episodic Pain Syndrome (FEPS) in Japan and Proposal of Clinical Diagnostic Criteria"

_ijms, 2024, doi:10.3390/ijms25136832_

Round 1

Reviewer 1 Report

Comments and Suggestions for Authors

Patients throughout Japan with familial episodic pain syndrome (FEPS)  underwent sequencing of SCN11A, SCN10A, and SCN9A genes for all variants. About 30% had Pathogenic or Likely Pathogenic variants per ACMGG criteria. Clinical information on these patients is presented, and used to validate the diagnostic criteria for FEPS.

This paper provides useful information on the clinical presentation of FEPS. In particular, it validates the earlier-proposed diagnostic criteria.

My only major feedback is that the Conclusion implies that SCN11A is the major cause of FEPS.

"Our data suggest that VGSCs abnormalities constitute part of the pathogenesis of FEPS and that SCN11A is the main cause of FEPS, with a typical course of FEPS."

However, counting VUSs, only 35% of FEPS patients have a variant in the 3 genes. I suggest rewording as such:

"Our data suggest that VGSC variants constitute a significant minority of the pathogenesis of FEPS. Among the 3 VGSC genes, SCN11A is the most common and the disease course is predominately typical, including early age of onset. However, including VUSs, only 35% of FEPS patients have a variant in those 3 genes. Thus, in the majority of cases the genetic predisposition of FEPS has not yet been established, and there may be other genes common in this disorder."

Minor point: Add the gene names (e.g., "SCN11A") to Figure 2, not just to the legend.

Comments on the Quality of English Language

In general, the paper is well-written but many sentences are difficult to read due to language and typos, and moderate English editing is needed. 

Reviewer 2 Report

Comments and Suggestions for Authors

FEPS is a rare autosomal dominant disorder whose genetic basis is already known.  With this manuscript, the authors describe genetic studies conducted in a large series of 212 Japanese patients with FEPS. Although this was a nationwide recruitment, such extensive genetic screening in FEPS patients had never been performed before.

Genetic testing was conducted by analyzing three of the four genes involved in the disease, namely SCN11A, SCN10A and SCN9A. The authors did not analyze the TRPA1 gene, which is reported in the literature to be involved in only one Colombian family. They found that 40 pathogenic and probably pathogenic variants were present in 30.2% of patients (64). Including VUS, the percentage of positive patients reached 35 percent. Some mutations were recurrent in the cohort. The authors also found that among patients with a positive genetic diagnosis, 89% of those carrying SCN11A met the recruitment criteria, divided into primary and secondary.

Major concerns

Why did the authors not analyze the TRPA1 gene as well? Since the genetic studies were conducted by WES, it would have been interesting to take a look at the TRPA1 gene as well, and even a negative result would have been interesting to report.

About 65% of patients with FEPS had no mutations. Looking at Table 2, there seems to be no difference in the clinical diagnostic criteria compared with the positive patients. This is my impression, but it needs to be confirmed or rejected. The authors did not report this result in the results section or comment on it in the discussion.  This is an important piece of information. If there were no differences in clinical criteria between patients with positive or negative genetic testing, hence what the Authors stated in the conclusion section, namely, "Most patients with pathogenic or probably pathogenic variants of SCN11A, SCN10A, and SCN9A met the clinical diagnostic criteria proposed in this study. This suggests that the criteria may be useful for the identification and diagnosis of patients with this disease" becomes less stringent. On the other hand, it might suggest that other FEPS disease genes could picked up in this large cohort by WES.

Minor

In paragraph 2.1 of the results section: why don't you describe at the beginning the clinical criteria for patient inclusion, the ones shown in Table 3? I think stating the inclusion criteria in this paragraph helps to better understand the purpose of your study.

Were the 212 families unrelated? If yes, a founder effect for recurrent mutations should be suspected.

Why did you use the term "possible" for patients with FEPS ? Do you mean that clinical criteria may not be sufficient to make a diagnosis of FEPS?

Is it possible to add a column to Table 1 to indicate which mutations are novel, and for those already reported what the respective references are?

In Table 3, the definition of definite diagnosis and probable diagnosis should be removed. It is a bit confusing to find these definitions in a table reporting the clinical diagnostic criteria for recruiting patients with FEPS. These definitions should be left only in the text.

In this regard, I also found the sentences in paragraph 4.3, lines 407-412, of the Materials and Methods section a bit confusing. What I do not find relevant is that the inclusion criteria for patient recruitment include the presence of pathogenic and probably pathogenic variants in VGSC genes. However, as far as I understand from the purpose of the study, the genetic data are the results of your study and the definition of definitive diagnosis of FSP is the conclusion of your study. I suggest moving the sentences to the discussion or conclusion.

Reviewer 3 Report

Comments and Suggestions for Authors

This manuscript is difficult to follow because it is somewhat incoherent, and the English should be greatly polished.

For one thing, the abstract should convey what exactly the authors have done and what they propose should be done later. The authors recruited 212 FEPS patients in Japan based on clinical diagnostic criteria. Line 72: They show that up to 30% of these patients harbored pathogenic variants in VGSC genes. Conversely, among patients with VGSC variants, more than 50% met the clinical criteria for FEPS. The authors conclude that these results suggest "the validity of the criterion” (line 77) – the latter should presumably be “validity of the clinical criteria”.

The main message seems to be that presence of VGSC variants confirms a clinical diagnosis of FEPS.

Line 78: “genetically diagnosed patients” should probably be “patients with underlying pathogenic variants”.

On line 122, the authors propose to perform genetic testing for VGSC variants on large numbers of FEPS patients. Is any of this new? It has been known that FEPS is caused by VGSC variants, so the authors should clarify what is new here.

Line 138: What portion of the 213 patients in this study have previously been described in the authors’ publications (refs 8 and 10)?

Line 156: What are “possible patients with FEPS”? Should this mean “patients with possible diagnosis of FEPS”?

Line 190: This title is unclear. Should “form” be “from”? Please reformulate clearly.

Line 210: What are “genetic variants with FEPS”?

Line 218: “patients with FEPS” should be replaced by “FEPS patients”

Line 240: “seemed to show be limited” is unclear.

Line 254: What does “more than several times” mean? More than 10 times? 20 times?

Line 318: What does “remain unclear the channel properties” mean?

Comments on the Quality of English Language

The English requires major polishing.

Round 2

Reviewer 2 Report

Comments and Suggestions for Authors

The authors adequately addressed my concerns

Reviewer 3 Report

Comments and Suggestions for Authors

The authors have made a major effort to update their manuscript. It now reads much better, and the aims of the study are easy to understand.